# Controlled Release of Phycocyanin in Simulated Gastrointestinal Conditions Using Alginate-Agavins-Polysaccharide Beads

**DOI:** 10.3390/foods12173272

**Published:** 2023-08-31

**Authors:** Alejandro Londoño-Moreno, Zayra Mundo-Franco, Margarita Franco-Colin, Carolina Buitrago-Arias, Martha Lucía Arenas-Ocampo, Antonio Ruperto Jiménez-Aparicio, Edgar Cano-Europa, Brenda Hildeliza Camacho-Díaz

**Affiliations:** 1Laboratorio de Metabolismo I, Departamento de Fisiología, Escuela Nacional de Ciencias Biológicas, Instituto Politécnico Nacional, Ciudad de México 07738, Mexico; alejo96m@gmail.com (A.L.-M.); zayra.mundofranco14@gmail.com (Z.M.-F.); mfrancoc@ipn.mx (M.F.-C.); 2Departamento de Biotecnología, Centro de Desarrollo de Productos Bióticos, Instituto Politécnico Nacional, Carretera Yautepec-Jojutla, Km 6, Calle CEPROBI No. 8, Morelos C.P. 62731, Mexico; cbuitragoarias@gmail.com (C.B.-A.); mlarenas@ipn.mx (M.L.A.-O.); aaparici@ipn.mx (A.R.J.-A.)

**Keywords:** biopolymers, diffusion, encapsulation, proteins, retention, stability

## Abstract

C-phycocyanin (CPC) is an antioxidant protein that, when purified, is photosensitive and can be affected by environmental and gastrointestinal conditions. This can impact its biological activity, requiring an increase in the effective amount to achieve a therapeutic effect. Therefore, the aim of this study was to develop a microencapsulate of a complex matrix, as a strategy to protect and establish a matrix for the controlled release of CPC based on polysaccharides such as agavins (AGV) using ionic gelation. Four matrices were formulated: M1 (alginate: ALG), M2 (ALG and AGV), M3 (ALG, AGV, and κ-carrageenan: CGN), and M4 (ALG, AGV, CGN, and carboxymethylcellulose: CMC) with increasing concentrations of CPC. The retention and diffusion capacities of C-phycocyanin provided by each matrix were evaluated, as well as their stability under simulated gastrointestinal conditions. The results showed that the encapsulation efficiency of the matrix-type encapsulates with complex composites increased as more components were added to the mixtures. CMC increased the retention due to the hydrophobicity that it provides by being in the polysaccharide matrix; CGN enabled the controlled diffusive release; and AGV provided protection of the CPC beads under simulated gastrointestinal conditions. Therefore, matrix M4 exhibited an encapsulation efficiency for CPC of 98% and a bioaccessibility of 10.65 ± 0.65% after the passage of encapsulates through in vitro digestion.

## 1. Introduction

C-phycocyanin (CPC) is a water-soluble, nutraceutical, hexameric protein that exhibits an intense blue color and absorbs light at a wavelength of 620 nm. The monomer of this protein is composed of two polypeptide chains, α and β, with molecular weights ranging from 18 to 20 kDa, and three phycocyanobilins (PCBs) as prosthetic groups (nonamino acid components of the protein). In the α subunit, the PCBs are located at Cys-84, while in the β subunit, the PCB is positioned between Cys-84 and Cys-155 [1,2,3]. Several studies have reported the bioactive properties of CPC, including antioxidant, anti-inflammatory, anticancer, cytoprotective, antihypertensive, and immunomodulatory activities, and potential modulation of intestinal microbiota, promoting gut health [4,5,6,7].

The physiological environment poses a challenge to the stability of nutraceuticals, such as CPC, as they undergo unfolding and loss of quaternary structure upon metabolism due to acidic pH levels. Gastric enzymes further hydrolyze peptide bonds, releasing amino acids and chromopeptides containing PCB, which affects their bioactivity and requires high doses to achieve therapeutic effects, increasing the likelihood of escalating the final cost of a CPC-based biopharmaceutical formulation. Additionally, factors such as light, oxygen, pH changes, and temperature have been reported to affect the stability and bioactivity of purified CPC [8]. Therefore, encapsulation has been proposed as a strategy to protect and provide enhanced stability to CPC.

Ionic gelation encapsulation is a method that provides protection to bioactive materials through hydrophilic or hydrophobic networks created by the biopolymers constituting the encapsulated wall material. These polymers should have the ability to form emulsions with the compound to be encapsulated, allow the release of the bioactive compound, not react with the material contained in the core, and withstand the encapsulation method [9,10]. Alginate has been widely used in ionic gelation due to its ease of use, low cost, and classification as a safe food ingredient [11,12,13]. Extensive research has showcased the potential of sodium alginate as a practical choice for protecting bioactive compounds. This is primarily due to its ability to retain its structural integrity even at reduced pH levels, such as the physiological pH levels found in the stomach. It offers a prospective delivery material to efficiently transport bioactive components through the gastrointestinal tract while maintaining their nutritional attributes [13]. However, a disadvantage of alginate is that it creates porous matrices, making it difficult to control the release of encapsulated compounds. Therefore, it has been suggested to use alginate in combination with other biopolymers to improve the physical and mechanical properties of the gel [14]. An example of this is agavins, agave reserve polysaccharides that structurally consist of branched fructose chains with terminal glucose, which, due to their β 2-1 and β 2-6 linkages, provide functional properties that enhance the efficiency of encapsulation systems and confer a prebiotic bioactive property to the encapsulating matrices, modulating the gut microbiota and the production of beneficial compounds for gastrointestinal health, such as short-chain fatty acids (SCFAs) [15,16]. The choice of the constituent components of the outer layer in encapsulation is of paramount importance, as the characteristics of these materials can influence the retention (molecules not released into the medium), diffusion (controlled release due to concentration gradients), protection (preservation of molecular function) from environmental or gastrointestinal conditions, and establishment of a matrix that allows dosing and release of the desired bioactive compound. Therefore, the aim of this study was to develop a microencapsulate of a complex matrix using polysaccharides with different chemical structures and functional properties for controlled release and dosing under simulated gastrointestinal conditions to improve the stability and bioaccessibility of nutraceuticals, such as CPC.

## 2. Materials and Methods

### 2.1. Materials

C-phycocyanin (CPC) was purified from the cyanobacteria *Arthrospira maxima* cultured in Zarrouk medium, at 21 ± 2 °C, with constant aeration provided by an air pump, under white LED illumination (24 W, 3000 Lx) and a 12/12 h light/dark cycle, as previously described [6,17]. The semidry biomass of *A. maxima* was resuspended in distilled water and subjected to freezing and thawing cycles. Subsequently, 10 centrifugation cycles were carried out at 21,400× *g* for 10 min at 4 °C to eliminate cell debris and obtain an extract rich in phycobiliprotein. This concentrate of phycobiliproteins was injected into a Sephadex^®^ G-100 column previously equilibrated with 10 mM PBS (pH 7.4). The bluish fractions were obtained and precipitated with a saturated solution of (NH_4_)_2_SO_4_ at 4 °C for 24 h in complete darkness, then subjected to dialysis and lyophilized. The agavins (AGV) of *Agave angustifolia* Haw were obtained in powder form following the Mexican patent MX/a/2015/016512 (modular system and process for obtaining different products from agave fructans). Sodium alginate (ALG) REASOL^®^, molecular weight 216 g/mol, purity (95–100%), κ-carrageenan (CGN) molecular weight 789 g/mol and carboxymethylcellulose (CMC) molecular weight 262 g/mol, food grade. The manipulation of CPC for encapsulation and quantification was carried out in darkness.

### 2.2. Encapsulation of CPC by Ionic Gelation

For the process of encapsulation through ionic gelation, sixteen formulations were prepared, as described in Table 1. Four solutions with different polysaccharide blends (denoted as M1, M2, M3, and M4) were used to encapsulate four increasing concentrations of CPC (10, 12, 15, and 20 mg/mL). The control encapsulations contained only the polysaccharide blends without CPC.

As a diluent, a 10 mM pH 7.4 phosphate buffer was employed. This buffer was heated to an approximate temperature of 50 °C to dissolve the alginate (ALG), κ-carrageenan (CGN), and/or carboxymethyl cellulose (CMC), as appropriate. Subsequently, the agavins (AGV) were added at room temperature to prevent caramelization. Once the polysaccharides were dissolved, the C-phycocyanin was added at 4 °C, while stirring until a homogeneous mixture was achieved. The preparations were placed in sterile, airtight containers and stored at 4 °C for 12 h in darkness to eliminate excess bubbles generated during the reagent dilution process. Finally, the formulations were passed through a peristaltic pump (INTLLAB), which directed the constant dripping of each mixture into a 0.2 M CaCl_2_ solution, allowing for the gelation of the matrices. The tubing was washed and disinfected with 70% alcohol before and after use.

### 2.3. Experimental Optimization Design for Wall Materials

Based on previous experimental tests, experimental design (DOE) was conducted using Minitab^®^ software v.18. The factors considered were the concentration of the biopolymers in the matrix, agavins, κ-carrageenan, and carboxymethylcellulose, with the intention of maintaining a total of 6% solids contributed by the polymers in the matrices for encapsulating four concentrations of CFC (10, 12, 15, and 20 mg/mL). From these evaluated factors, the established responses were defined as encapsulation efficiency (*EE*%), diffusion into the medium, and retention of CFC for each matrix. The concentration of alginate was not considered, as it was consistently maintained at 1% within the designed matrices [16].

### 2.4. Encapsulation Efficiency (EE%)

The encapsulation efficiency (%) was determined by obtaining the difference between the total *CPC* and the *CPC* in the encapsulating medium divided by the total amount of *CPC*, per hundred, as shown in Equation (1).
(1)$$(EE\%)=\frac{TCPC-CPC}{TCPC}\times 100$$
where *TCPC* is the total C-phycocyanin before the encapsulation process and *CPC* is the amount of nonencapsulated C-phycocyanin found in the CaCl_2_ solution.

The C-phycocyanin concentration with the highest encapsulation efficiency was studied by Fourier transform infrared spectroscopy (FT-IR) using a spectrophotometer (IRAffinity-1, Shimadzu, Kyoto, Japan) with an ATR (attenuated total reflection) accessory with a zinc selenide crystal. For each spectrum, an average of 16 scans was recorded, with a resolution of 4 cm^−1^ and scan 60 in the range of 400–4000 cm^−1^.

### 2.5. Stability Tests on CPC Encapsulation

The stability of the encapsulated CPC was determined by the retention and diffusion of CPC in the beads. After the encapsulation process by ionic gelation for each formulated matrix (Table 1), approximately 100 mg of beads were weighed and broken down in a 10% sodium citrate solution (Na_3_C_6_H_5_O_7_) at 0, 30, 60, 90, and 120 min after the encapsulation process. In parallel, the amount of C-phycocyanin that diffused into the CaCl_2_ medium was measured at 0, 10, 20, 30, 60, 90, and 120 min after the encapsulation process. A timeframe of 120 min approximates the duration of the digestion process of a food item through the oral and gastric phases and enables the metabolism of CPC at the intestinal level. This was considered, given the requirement for minimal CPC release prior to reaching the small intestine. 

CPC quantification was performed to measure stability with a Multiscan Go^®^ microplate spectrophotometer (Thermo Scientific, Co., Waltham, MA, USA) at 620 nm, in triplicate. Previously, a standard curve was made for each formulated matrix.

### 2.6. Morphometric Characterization of the Encapsulates

The physical characterization of the matrix-type encapsulates was performed using an epifluorescence microscope (EFM) with an attached digital camera (Nikon 50i) and NIS Element software (versión 2.30) to capture the microphotographs (exposure of 33 ms in bright field and exposure of 300 ms in epifluorescence under green laser, stored in *.TIFF color format). The micrographs were analyzed using the open-source software ImageJ (v1.44p). The images were converted to grayscale (8 bits), binarized (black and white) using the “Threshold” function, and the grayscale range was manually adjusted. Based on these beads’ images, size parameters (area, Feret diameter, and perimeter), as well as shape parameters (circularity and solidity) were analyzed. These morphometric parameters were evaluated to identify which encapsulates were significantly influenced by the presence of different wall materials (alginate, agavins, κ-carrageenan, and carboxymethyl cellulose). This analysis was performed for all formulations encapsulated with 20 mg of CPC, as well as the respective control encapsulate (without added CPC). A principal component analysis (PCA) of the size and shape parameters was performed using Minitab^®^ software v.18.

### 2.7. In Vitro Digestion of Matrix-Type Encapsulates

The in vitro digestion process, which consists of three simulated phases (simulated oral phase, SOP; simulated gastric phase, SGP; and simulated intestinal phase, SIP), was carried out following the INFOGEST protocol published in 2019 by Brodkorb et al. [18]. Initially, simulated gastrointestinal fluid solutions were prepared from stock solutions: salivary (SSF), gastric (SGF,) and intestinal (SIF) fluids. For SSF, α-amylase (Sigma-10700 41.2 U/mg) was added at a concentration of 10 mg/mL; pepsin (Sigma-P7000 > 250 units/mg solid) 20 mg/mL and lipase (Sigma-L3126) at 100 mg/mL were added for SGF; and for SIF, pancreatin (Sigma-P1750 100 g) 133.3 mg/mL and bile salts (OXOID LP0055) 200 mg/mL were added. For the SOP, 0.2 mL of SSF (pH 7.0) was added to 200 mg of beads in test tubes and incubated for 2 min (sampling at 0 and 2 min); then, for the SGP, SSF and SGF (pH 3.0) in a 1:1 (*v*/*v*) ratio were added to the tubes and incubated for a period of 2 h (sampling at 0, 1, and 2 h). Finally, the simulation of the SIP was achieved by adding the SIF (pH 7.0) in a 1:1 (*v*/*v*) ratio to the tubes containing SSF and SGF and incubating them for a period of 2 h (sampling at 0, 1, and 2 h). The tubes were incubated at 37 °C with shaking. NaOH (5 M) and HCl (5 M) solutions were used for pH adjustment. *CPC* quantification was performed at 620 nm in triplicate. Additionally, the bioaccessibility (B %) of the *CPC* in the matrix encapsulates was determined at the initial point of the SIP (130 min) using Equation (2) [19].
(2)$$B\;(\%)=\frac{CPC\;content\;(mg\;CPC/g\;matrix\;type\;encapsulates)}{Initial\;CPC\;content\;(mg\;CPC/g\;matrix\;type\;encapsulates)\;\;}\times 100$$
where the *CPC content* is the amount of C-phycocyanin present at the initial time of the simulated intestinal phase (130 min) and the initial *CPC* is the C-phycocyanin content at the beginning of the in vitro digestion process. In addition, the release of C-phycocyanin in the simulation in vitro was analyzed according to the Korsmeyer–Peppas model (Equation (3)), and the parameters *n* and *k* were determined [20].
(3)$$\frac{Mt}{M\infty }=k_{p}t^{n}$$

In this equation, *Mt/M∞* represents the fractional permeated compound, *k_p_* is a system-specific constant, and *n* corresponds to the transport mechanism. For spherical particles, *n* ≤ 0.43 for Fickian diffusion (Case I transport); 0.43 ≤ n ≤ 0.85 for non-Fickian diffusion or swelling; *n* ≥ 0.85 for a Case II transport; and *n* > 1 for a Super Case II transport.

The zero-order model (Equation (4)) considers the released fraction to be independent of the initial concentration [20].
(4)$$\frac{Mt}{M\infty }=kt$$

The rate of a first-order (Equation (5)) release model is dependent on the released concentration [20].
ln *Mt* = ln *M∞* − *kt*(5)

At the simulated gastrointestinal digestion sampling points, photomicrographs of the encapsulates were obtained through bright-field and epifluorescence microscopy (EFM) (capture conditions described above). In addition, photomicrographs were captured through confocal laser scanning microscopy (CLSM) using CLSM software (Carl Zeiss LSM800), ZEN (Zeiss Efficient Navigation) version 2.6 Blue Edition. The micrographs were acquired with a 20× and 40× apochromatic objective, with numerical apertures of 0.8 and 1.3, respectively. A 488 nm laser was used with excitation at 495 nm and emission at 519 nm and a 561 nm laser with excitation at 613 nm and emission at 629 nm; both lasers had 3% excitation with a pinhole aperture of 1.0 airy units (AU). All micrographs were stored in *.TIFF format, with a resolution of 1024 × 1024 pixels at 300 dpi and an environmental scanning electron microscope (ESEM) Carl Zeiss EVO LS 10 (Life Science, Oberkochen, Germany) with Zeiss Efficient Navigation Software (2.3 Blue Edition) at 250× optical magnification with backscattered electrons in the environmental mode. Three micrographs were taken for each type of bead (250× magnification).

### 2.8. Statistical Analysis

All values represent the mean ± SEM. A two-way ANOVA with Tukey’s post hoc test (*p* < 0.05) was performed. For data analysis and illustration, the SigmaPlot statistical package (v 12.0), GraphPad Prism software (v 8.0), and the Minitab Graphs package (Minitab^®^18.1) were used. 

## 3. Results

### 3.1. Experimental Optimization Design for Wall Materials

The contour plot (Figure 1) highlights the relevance of the polymers agavins (*AGV*), κ-carrageenan (*CGN*), and carboxymethylcellulose (*CMC*) in the encapsulation efficiency. The experimental design analysis showed that higher concentrations of *CGN* led to higher encapsulation efficiency, while the concentration of agavins was inversely proportional to the *EE*%, meaning that higher amounts of agavins resulted in lower efficiency. However, the modeling of retention and diffusion of *CPC* indicated that *CMC*, along with *AGV* and *CGN*, allowed for sustained release of the protein and reduced diffusion into the medium. Furthermore, the interaction of these components with alginate (not considered a variable, as its concentration remained constant at 1% in all formulations) contributed to the formation of the encapsulating matrix, ultimately improving both encapsulation efficiency and retention.

### 3.2. Encapsulation Efficiency (EE%)

The statistical analysis indicated significant differences both among matrices (M1, M2, M3, and M4) and among *CPC* concentrations (10, 12, 15, and 20 mg/mL). Likewise, the interaction analysis of both variables showed that for the formulation containing only alginate (M1), the highest encapsulation efficiency was 79.33 ± 0.31%, observed when formulating the matrix with a concentration of 15 mg/mL of *CPC*. Following this, an efficiency of 76.50 ± 0.31% was found when attempting to encapsulate matrix M1 with only alginate and 20 mg/mL of *CPC*. In the case of matrix M2, with alginate and agavins, the highest encapsulation efficiency found was 95.20 ± 0.44%, observed when formulating it with 10 mg/mL of *CPC*. It was also observed that increasing the phycocyanin concentration decreased the encapsulation efficiency. On the other hand, the ternary matrix (M3), with alginate, agavins, and κ-carrageenan, exhibited the highest values in terms of the matrix’s capacity to retain the protein, initially reaching a maximum encapsulation efficiency of 99.79 ± 0.31% when formulated with both 12 and 15 mg/mL of CPC. Finally, matrix M4, with carboxymethyl cellulose and the aforementioned polysaccharides, achieved an encapsulation efficiency above 98% when formulated with 20 mg/mL of CPC, representing the highest retained concentration of 163.97 ± 3.13 mg of CPC/g of matrix-type encapsulates.

Additionally, it can be observed in Figure 2 that all formulations showed a significant increase in protein encapsulation efficiency, regardless of the evaluated CPC concentration, compared to matrix M1, which served as the encapsulation process control by being formulated solely with alginate. Furthermore, formulations M3 and M4 generally exhibited an encapsulation efficiency above 96%. Significant differences (*p* < 0.05) were found among the concentrations of CPC (10, 12, 15, and 20 mg CPC/mL) and among the evaluated matrices (M1, M2, M3, and M4).

Fourier transform infrared spectroscopy (FT-IR) was employed to identify modifications in protein structures and explore the interactions between proteins and polysaccharides (Figure 3). In the region 1600–1700 cm^−1^, there is a band corresponding to the amide I for C=O stretching, which can be utilized to examine the secondary configuration of proteins responsible for the CPC antioxidant activity. The spectrum of M4 beads did not exhibit changes in that region, indicating the preservation of the secondary conformation of the CPC protein. CPC had characteristic peaks at 1651 cm^−1^ (C-O stretching vibration) and 1545 cm^−1^ (N–H bending vibration). In addition, the amide I, amide II, and amide III (1651 cm^−1^, 1542 cm^−1^, and 1392 cm^−1^, respectively) peaks have been slightly red-shifted and their intensities changed after reactions with ions or molecules, indicating changes in the secondary structure of C-phycocyanin. The band at 950–1000 cm^−1^ was correlated to the extension of C-O and C-C and the bending of C-H bonds [21].

### 3.3. Stability of the CPC Encapsulation

Within the design of the encapsulates as dosing systems to be administered orally (capsules, tablets, or others), it is important to monitor the protein concentration that each formulated matrix encapsulated could retain, aiming to achieve the maximum amount of C-phycocyanin retention in the minimum quantity of encapsulates. Additionally, understanding the effect of component additions to the blends and elucidating suitable times, concentrations, and materials that would enable the best encapsulation efficiency was crucial in minimizing protein loss from the encapsulates and maintaining constant bioactive concentrations. Furthermore, the composition of the appropriate wall materials should not only provide protection to phycocyanin from gastrointestinal conditions but also allow for sustained dosing to generate a therapeutic effect. 

Among all the formulated matrices, M1 (1% alginate) exhibited the lowest protein retention (Figure 4A). The CPC concentrations alone did not result in changes in retention (30–45%) at time zero of the encapsulation process. However, the interaction between CPC and time (min) revealed that, at extreme protein levels (in the case of this experiment, 10 or 20 mg CPC/mL), the minimum retention percentage (<15%) was achieved. On the other hand, the diffusion percentage (Figure 4E) was higher (>60%) when the matrix contained the lowest amount of CPC. When agavins were added as a component of the matrix (M2), an improvement in protein retention, determined by an increase in encapsulation efficiency (Figure 4B) and a lower percentage of CPC diffusion into the medium (Figure 4F), was observed (<20%, except for concentrations close to 20 mg CPC/mL with diffusion values between 20 and 40%). However, it was found that with a longer time (min), a lower percentage of CPC retention could be found in the capsules, with a loss of protein retention of approximately 45% at 60 min postencapsulation. Furthermore, it was observed that at lower CPC concentrations in the formulated matrix, a greater protein loss could occur, with the lowest retention percentage between 15 and 30% starting from 60 min, for formulations with 10 and 12 mg/mL of CPC. In other words, the addition of agavins to the matrix improved protein retention, and this retention percentage was lost to a lesser degree when more than 12 mg/mL of C-phycocyanin was added.

### 3.4. Morphometric Characterization of the Encapsulates

In the experimental development of encapsulation, it was qualitatively found that increasing the components in the matrices resulted in greater mechanical resistance in the matrix-type encapsulates, making them more solid to the touch. Figure 5 shows the increased size of the obtained encapsulates as components were added to the mixture and records the morphometric values (), indicating that the increase in biopolymers in the formulations led to an increase in size parameters (area, diameter, and perimeter). Furthermore, it was observed that the concentration of C-phycocyanin in the matrices also modified the morphometric values of the encapsulates obtained (Appendix A). Thus, matrix M4, containing alginate, agavins, κ-carrageenan, carboxymethylcellulose, and 20 mg of CPC/mL, exhibited the largest size and shape values. The interaction of the phycocyanin between the different wall materials used was in the following order: AGV> CGN > ALG > CMC; this corresponds to the strength of the bond and was promoted by the addition of each of the components to the matrix.

### 3.5. In Vitro Digestion of Matrix-Type Encapsulates

In general, it was observed that a CPC decrease in all samples during the gastric phase was due to the protein’s susceptibility to pepsin present in SGP, (Figure 6A,B), which was related to the greater release of the component in that phase of the simulated digestion process. The correlation between the released CPC concentration during the digestive process and the luminosity of the beads surface, evaluated by epifluorescence microscope (Appendix A), was observed in the following order: M4 > M3 > M2 > M1, with the degradation of CPC being later and at a lesser amount in M4.

During the in vitro digestion process, the M1 matrix, with the lowest retained CPC concentration, was not affected in the simulated oral phase (SOP) (Figure 7 M1-SOP). However, monitoring M1 in the simulated gastric phase (SGP) revealed a sustained decrease in the amount of CPC (mg CPC/g encapsulated matrix) (Figure 6A). Additionally, microscopy analysis showed a decrease in the blue pigment quantity and fluorescence intensity captured by confocal laser scanning microscopy (CLSM) (Figure 7 M1-SGP). Finally, at 130 min, the start of the simulated intestinal phase (SIP), a significant decrease in both concentration and luminosity in the beads was observed (Figure 6A,B), along with a near absence of fluorescence captured through multiepifluorescence (MEF) and CLSM (Figure 7 M1-SIP). Furthermore, environmental scanning electron microscopy (ESEM) photomicrographs revealed progressive damage to the surface of M1 encapsulates, characterized by increased striations. This indicates matrix wear and suggests that capsules formulated solely with alginate released the minimal protein content and facilitated the rapid action of proteolytic enzymes, resulting in limited retention and protection of the CPC proteins.

The addition of agavins to the encapsulating matrix of C-phycocyanin allowed for a progressive release of the protein from SOP to SIP. The M2 matrix exhibited protein aggregates that were visible under CLSM fluorescence (Figure 7 M2-CLSM) and, although decreasing in quantity, remained preserved during the simulated digestion process despite the observed matrix wear on the capsule surface as seen in ESEM images (Figure 7 M2-ESEM). In summary, M2 demonstrated better mechanical and structural resistance to in vitro conditions, providing greater protection against enzymatic attack compared to the M1 matrix (Figure 6). This was further evidenced by the higher preservation of luminosity and fluorescence related to CPC.

The increase in matrix components, including alginate, agavins, κ-carrageenan, and C-phycocyanin, resulted in enhanced CPC protection during digestion. The protein was dispersed more homogeneously within the capsule upon the addition of κ-carrageenan (Figure 7 M3-CLSM). This tertiary matrix (M3) reduced capsule permeability, protecting the protein from pH changes and enzymatic activity under simulated gastrointestinal conditions (Figure 6). Moreover, it exhibited greater structural stability on the surface of the encapsulate, with fewer signs of cracking observed at the end of SIP (Figure 7 M3-ESEM).

Lastly, the addition of CMC in the M4 matrix led to an increase in encapsulate area, as evident from the convolutions on the surface captured by environmental scanning electron microscopy (Figure 7 M4-ESEM). This was associated with enhanced CPC retention capacity and increased available contact surface area. Furthermore, M4 effectively preserved CPC during simulated gastrointestinal conditions, allowing for the gradual release of the protein and establishing a dosing and protection system for CPC under digestion-simulated conditions (Figure 6 and Figure 7 M4).

## 4. Discussion

The encapsulation of CPC using alginate as the wall material in ionic gelation has been reported. Yan et al. [11] found that when using 1% alginate, the encapsulation efficiency (EE) reached a value of 30.69 ± 2.34, and by increasing the ALG concentrations to 3.5%, a maximum efficiency of 62.66 ± 0.72% was achieved. Additionally, Hadiyanto et al. [8] indicated that EE and phycocyanin retention were directly proportional to the alginate concentrations. They found that 1.5% alginate resulted in a minimum encapsulation efficiency of 53.53 ± 0.61%, and increasing the alginate concentration to 2.5% only led to a slight increase in efficiency to 71.76 ± 0.13. These EE values are similar to those found for the M1 matrix, formulated with 10 and 12 mg/mL of CPC, where the efficiency reached 70%. However, increasing the CPC concentration on the matrix resulted in a 10% increase in encapsulation efficiency, reaching nearly 80%. It has been suggested that an electrostatic interaction is established between phycocyanin and alginate during the formation of the beads.

The amount of alginate is proportional to the number of polymer chains available to form the wall material, thereby increasing the coating and hardness of the capsule. Additionally, the influence of CPC on encapsulation efficiency should be considered [8,11,22,23,24]. Regarding the influence of agavins on encapsulation efficiency, an increase in EE was observed in the case of the M2 matrix. Chávez-Falcón et al. [16] reported that a combination of agavins and proteins (such as whey protein) mixed in a polymer matrix with sodium alginate significantly increased the EE, reaching values between 94 and 97% compared to alginate alone. The authors attributed this behavior to the heterogeneity in the internal structure (entropy and fractal dimension) provided by agavins, which improved the crosslinking of the polymer network and allowed better retention of the component (in their case, probiotic cells).

κ-carrageenan has also been used in CPC encapsulation processes, and it has been found that the use of this polysaccharide alone resulted in an encapsulation efficiency (EE) of 68.66 ± 2.45% [8]. In the experiment conducted in this research, the addition of kappa-carrageenan in the M3 encapsulating matrix significantly impacted the diffusion of CPC into the CaCl2 crosslinking solution, reducing it to below 20% and improving the CPC retention observed when agavins were added to the formulated mixtures (M2). Lei et al. [25] studied mixtures of CPC and increasing concentrations of kappa-carrageenan, finding that the strongest interaction occurred through disulfide bonds, followed by hydrophobic and electrostatic forces, with a weaker interaction through hydrogen bonding. These interactions were important for the formation of CPC and CGN-based gels, as they resulted in a denser, less porous network with reduced hydrophilicity, thereby improving the retention of the hydrophilic CPC protein.

On the other hand, Agarwal et al. [26] employed calcium alginate and CMC as the encapsulating matrix through ionic gelation and found that increasing CMC concentration led to a larger size and roughness on the surface of the beads. The polyelectrolytic natures of CMC and ALG are known to cause pH-dependent swelling that is attributed to the presence of negatively charged carboxyl groups in the polymer’s main chain. At acidic pH levels, the carboxylic acid groups remain undissociated, resulting in no net charge development within the polymer network. When exposed to an alkaline pH, such as that of the intestine, the carboxylic acid group converts to negatively charged carboxylate ions, leading to electrostatic repulsion between different polymer chains. This, in turn, causes the polymer network to swell, modulating the release of the bioactive material and creating a carrier and dosing system that exhibit greater mucoadhesion at alkaline pH levels compared to acidic conditions. This increases the retention time of the compound, providing an opportunity for complete degradation of the beads and thus sustained release [27,28]. The addition of CMC into the M4 matrix had a positive effect on the retention parameters of CPC. Furthermore, the increased surface area not only directly relates to the matrix’s ability to release CPC protein under intestinal conditions but also provides a larger interaction surface available for the microbiota to ferment components such as agavins, enabling the release of CPC at the intestinal level from the encapsulating matrix.

Different studies have shown that alginate is an ideal biopolymer to obtain high encapsulation efficiency and improve CPC stability under adverse environmental conditions. However, CPC in an aqueous solution is extremely unstable to light and heat; thus, obtaining microencapsulates in calcium chloride solutions using batch systems or continuously during the ionic gelation process depends on the wall material used, as a significant C-phycocyanin loss of at least 30% can be obtained in the best cases of encapsulation efficiency using, for example, chitosan [11], pectin [29], or gelatinized starch [21] in the formation of beads, coacervates, or water/water emulsions. Therefore, the results of encapsulation efficiency and release during storage in the calcium chloride solution in the present investigation were the highest values that have been reported so far for the encapsulation systems for CPC and various polysaccharides. According to Figure 3, at the highest CPC concentration of 20 mg/mL used in the mixture of quaternary complexes (alginate–agavins–carrageenan and carboxymethylcellulose), the highest retention values (75–99%) and the lowest release under storage conditions (1–20%) were obtained during the formation of microspheres in calcium chloride solution.

The solubility tests that Lei [25] performed in five different solvents clarified the role of protein-polysaccharide interactions in the formation of the gel network, in addition to identifying the differences between electrostatic interactions, hydrophobic interactions, hydrogen bonding, and disulfide bonding. It was possible to demonstrate these interactions for each wall material through FT-IR spectroscopy (Figure 3). The agavins favored the hydrogen bonds and therefore the degree of crosslinking of the microencapsulation, conferring greater size, greater irregularity, and greater solubilization of the CPC in the salivary medium.

Carrageenan, as already reported in other previous research papers such as Lei et al. [25] and Hadiyanto et al. [8], in the concentrations in which it was used, mainly allowed the formation of disulfide interactions within electrostatic interactions, which resulted in a better dispersion of the CPC in the matrix-type beads and therefore a greater retention of the component. This, in turn, resulted in the controlled release of CPC into the small intestine, with the highest % bioaccessibility in the intestinal phase (19%) of the CPC reported so far in previous works under in vitro gastrointestinal conditions for sodium alginate in phycocyanin emulsions [29] thanks to the interaction of agavins and carrageenan. Finally, carboxymethylcellulose, due to its nature of origin, gave rise to a hydrophobic interaction that allowed, in the form of a complex with the other components, a decrease in affinity of CPC with the aqueous medium and therefore decreased rates of diffusion of CPC in salivation (kp = 1.9900), gastric (kp = 0.1558), and intestinal (kp = 0.5357) conditions. Similar values were recently obtained by Azaza et al. [30] in chitosan/collagen-based hydrogels using the same Korsmeyer–Peppas release kinetics model (n = 1.36–1.61) with n > 1.0 for Case II transport (polymer erosion) in SOP and for SGP and SIP n < 0.4 diffusive.

The correlation coefficient (R^2^) values of all types of beads (Appendix A) did not follow zero order or first order, but the correlation coefficient (R^2^ = 0.9980) values were close to the Korsmeyer–Peppas model. The higher correlation coefficient for the release kinetics of CPC with the three diffusion models was for M4 beads. Furthermore, a better fit with the zero-order model was observed for the intestinal phase (SIP) and with the first-order and Korsmeyer–Peppas models for the gastric phase (SGP).

In addition, in our case, it was possible to preserve the structure of the beads until the end of the intestinal phase and therefore ensure that part of the phycocyanin reaches the colon, where its health benefits can be exploited, similar to the system of Wen et al. [31], who obtained tripolyphosphate nanofibers as a phycocyanin carrier to treat colon cancer. 

## 5. Conclusions

The present investigation focused on the optimized design of beads by ionic gelation using different polysaccharides (alginate, agavins, carrageenan, and carboxymethylcellulose) that have been shown to individually improve the stability of the encapsulated compound in conditions such as aqueous solutions, pH changes, light, and high temperatures. The polysaccharides in a complex matrix can interact synergistically, providing the best morphological, physical, chemical, and functional characteristics. The interaction between AGV (indigestible fiber) and CGN showed a high affinity for CPC that allowed its controlled release during gastric and intestinal digestion and its conservation until the colon; the addition of CMC (hydrophobic polysaccharide) resulted in the maximum CPC retention in the beads during the oral phase and compound stability until the colon. The results obtained in this study hold scientific significance by demonstrating a previously unreported possibility in the literature: employing a complex matrix as an alternative method for encapsulating C-phycocyanin through ionic gelation. This efficient, reproducible, and straightforward approach effectively preserved CPC under simulated gastrointestinal conditions, suggesting its potential application as a nutraceutical in preclinical research for drug development of cotreatments for human diseases.

## Figures and Tables

**Figure 1 foods-12-03272-f001:**
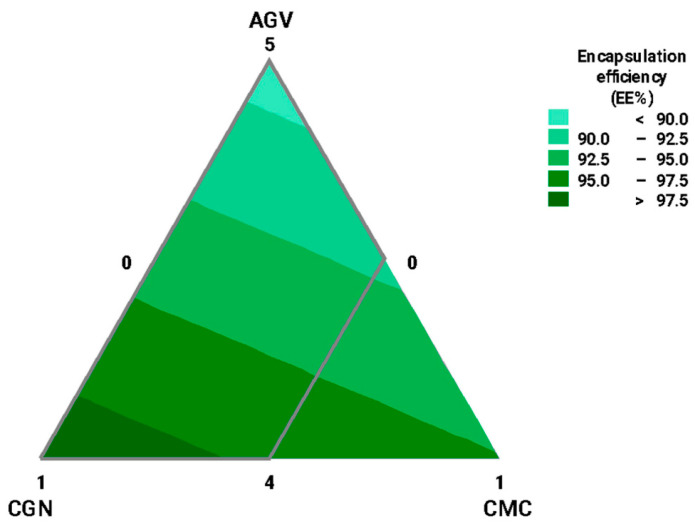
Design of experiment (*DOE*) for the optimization of the blend in relation to the encapsulation efficiency (*EE*%) and considering the amount of biopolymers as components of the wall material. Based on 6 measurements of encapsulation efficiency. *AGV*: agavins; *CGN*: κ-carrageenan; and *CMC*: carboxymethylcellulose. Alginate (*ALG*) was not considered, as its concentration remained constant between the formulated matrices (1%).

**Figure 2 foods-12-03272-f002:**
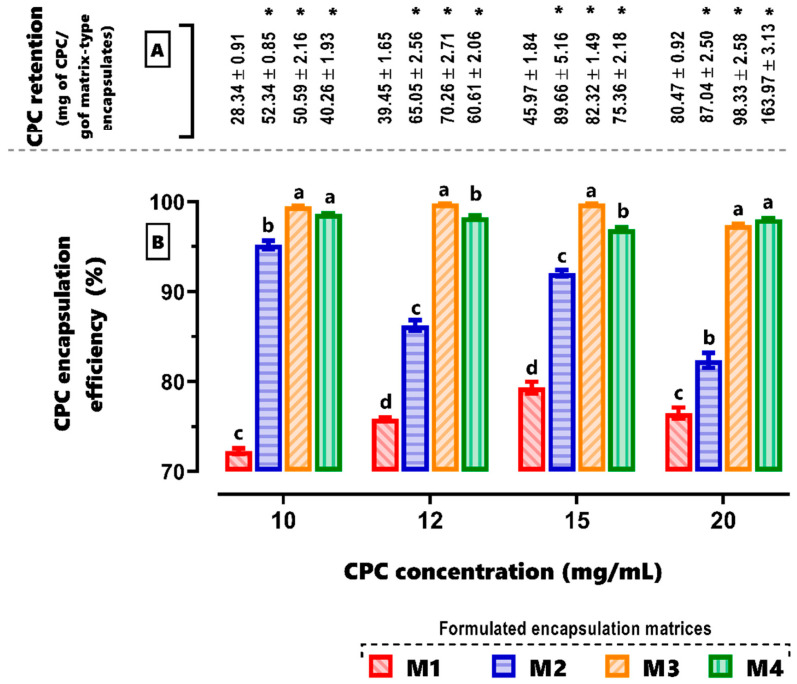
(**A**) Retention of CPC (mg of CPC/g of encapsulates) by each of the formulated matrices: M1, M2, M3, and M4. Values represent the mean of 6 measurements ± SEM. (*) *p* < 0.05. (**B**) Encapsulation efficiency (%) of the formulated matrices for the retention of increasing concentrations of CFC (10, 12, 15, and 20 mg/mL). Each bar represents the mean of 6 measurements ± SEM. Bars with different letters indicate significant differences, *p* < 0.05. Comparison among M1, M2, M3, and M4 for each CPC concentration. Two-way ANOVA with Tukey’s post hoc test. M1: alginate; M2: alginate and agavins; M3: alginate, agavins, and κ-carrageenan; M4: alginate, agavins, κ-carrageenan, and carboxymethyl cellulose. CPC: C-phycocyanin.

**Figure 3 foods-12-03272-f003:**
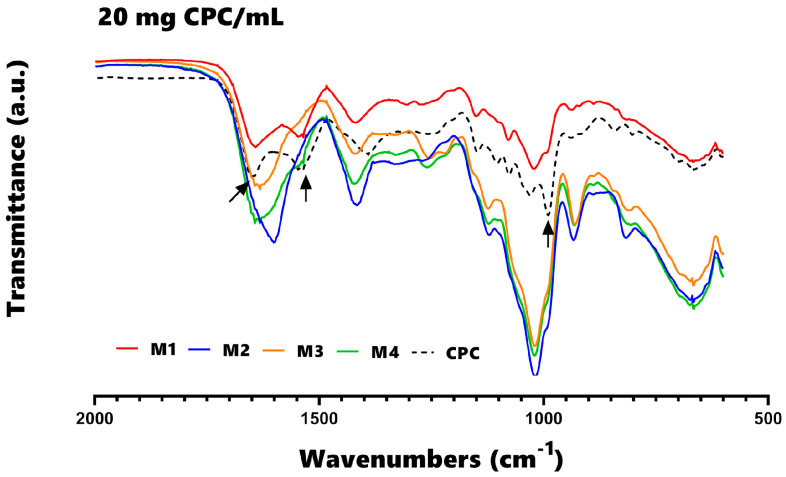
FT-IR spectra for each of the different matrix-type encapsulates formulated with 20 mg/mL of CPC. M1: alginate; M2: alginate and agavins; M3: alginate, agavins, and κ-carrageenan; M4: alginate, agavins, κ-carrageenan, and carboxymethyl cellulose. CPC: C-phycocyanin. The arrows represent (1651, 1542 and 1000-950 cm^−1^) peaks.

**Figure 4 foods-12-03272-f004:**
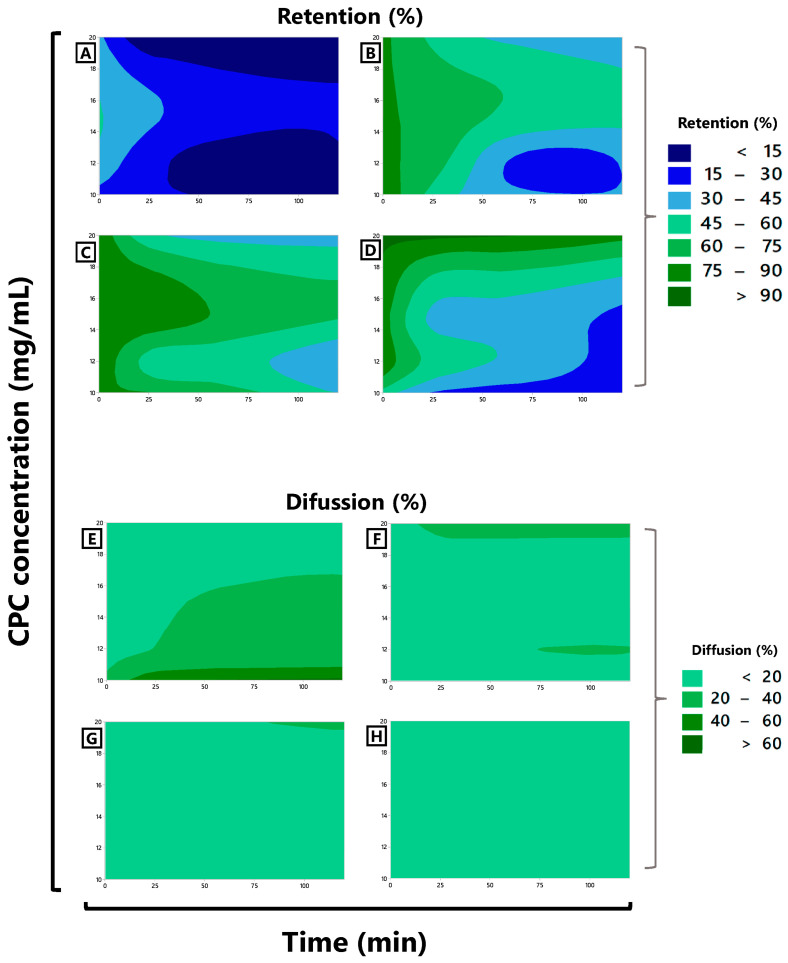
Contour plots for retention (%) versus CPC concentration (mg/mL) and time (min) for matrices M1 (**A**), M2 (**B**), M3 (**C**), and M4 (**D**), as well as diffusion (%) into CaCl_2_ solution versus CPC concentration (mg/mL) and time (min) for matrices M1 (**E**), M2 (**F**), M3 (**G**), and M4 (**H**) are shown. M1: alginate; M2: alginate and agavins; M3: alginate, agavins, and κ-carrageenan; M4: alginate, agavins, κ-carrageenan, and carboxymethylcellulose. CPC: C-phycocyanin. The CPC concentrations in the formulation of all matrices were 10, 12, 15, and 20 mg/mL.

**Figure 5 foods-12-03272-f005:**
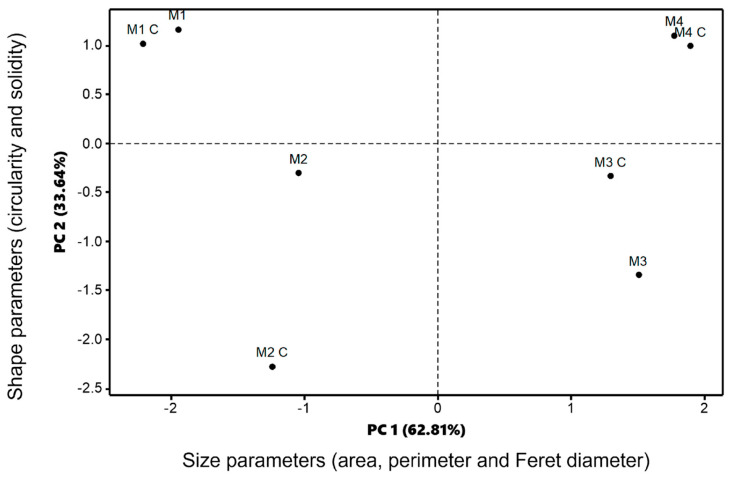
Principal component analysis (PCA) of the morphometric characteristics of matrix-type encapsulates formulated with 20 mg/mL CPC, associated with size parameters (area, perimeter, and Feret diameter) and shape parameters (circularity and solidity). The dots represent the measurement of 5 beads. Letter “C” represents the control matrix-type encapsulates without CPC added.

**Figure 6 foods-12-03272-f006:**
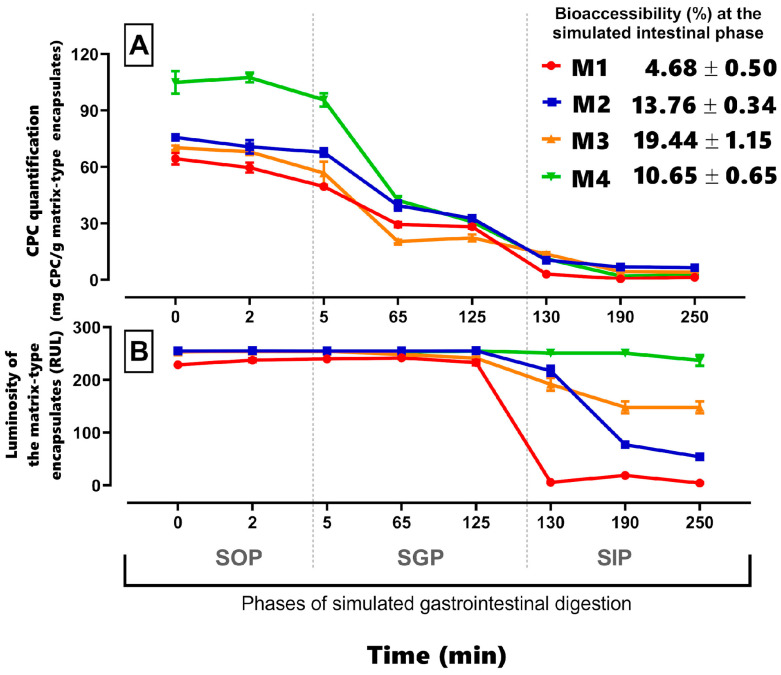
Simulated gastrointestinal digestion process of encapsulated matrix-type structures. (**A**) Quantification of CPC (mg CPC/g encapsulated matrix) and the potential bioavailability (%) of CPC in the simulated intestinal phase (SIP 130 min). Each data point represents the mean of 6 measurements ± SE. (**B**) Measurement of luminosity (URL) of the encapsulated matrix structures in a 200 × 200-pixel section in the central area of the photomicrographs obtained at each evaluated time point during the simulated digestion. Each data point represents the mean of 5 measurements ± SEM. M1: alginate; M2: alginate and agavins; M3: alginate, agavins, and κ-carrageenan; M4: alginate, agavins, κ-carrageenan, and carboxymethylcellulose. All formulations evaluated in the in vitro digestion process contained 20 mg/mL of C-phycocyanin (CPC). SOP: simulated oral phase; SGP: simulated gastric phase; SIP: simulated intestinal phase.

**Figure 7 foods-12-03272-f007:**
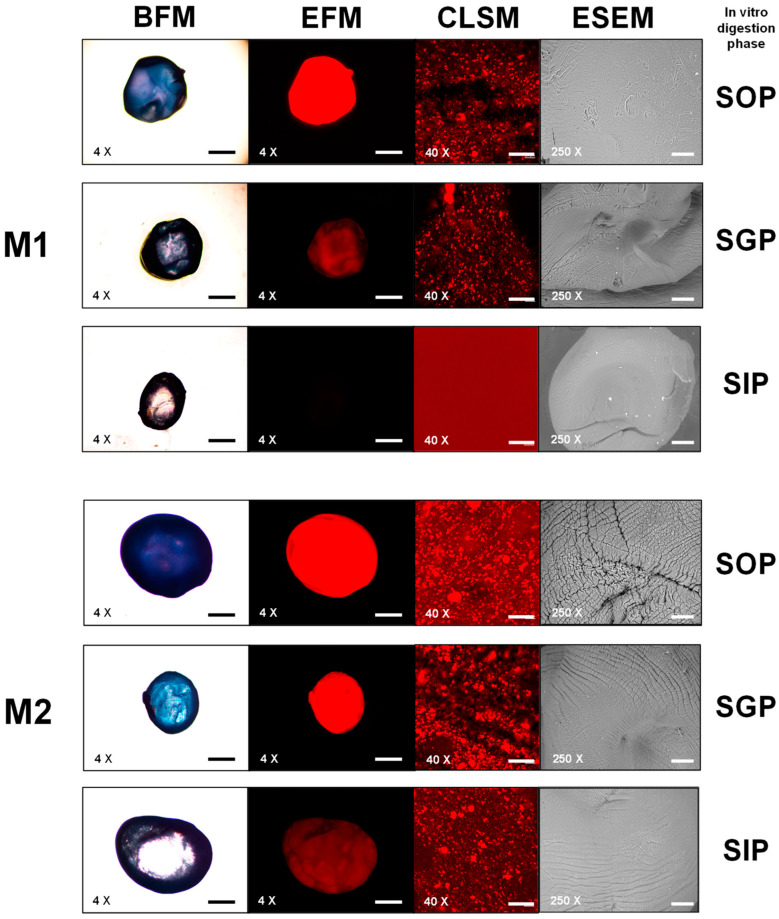
Micrographs of the encapsulates during the in vitro gastrointestinal digestion process through the simulated oral phase (SOP), simulated gastric phase (SGP), and simulated intestinal phase (SIP), captured using BFM: bright-field microscopy (magnification = 4×, scale bar 250 µm); EFM: epifluorescence microscopy (magnification = 4×, scale bar 250 µm); CLSM: confocal laser scanning microscopy (magnification = 40×, scale bar 20 µm); and ESEM: environmental scanning electron microscopy (magnification = 250×, scale bar 60 µm). M1: alginate; M2: alginate and agavins; M3: alginate, agavins, and κ-carrageenan; M4: alginate, agavins, κ-carrageenan, and carboxymethylcellulose. All formulations evaluated in the simulated gastrointestinal process contained 20 mg/mL of C-phycocyanin (CPC).

**Table 1 foods-12-03272-t001:** Nomenclature and composition of the formulations for the encapsulates.

Nomenclature Encapsulates	ALG	AGV	CGN	CMC	CPC
	(%)				(mg/mL)
M1	1				10, 12, 15 and 20
M2	1	5			
M3	1	4	1		
M4	1	4	0.5	0.5	

ALG: sodium alginate, AGV: agavins, CGN: κ-carrageenan, CMC: carboxymethylcellulose, and CPC: C-phycocyanin from *Arthrospira maxima*. All formulations were used to encapsulate 10, 12, 15, and 20 mg/mL of CPC.

## Data Availability

Not applicable.

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
