# Peer review of "Controlled Release of Phycocyanin in Simulated Gastrointestinal Conditions Using Alginate-Agavins-Polysaccharide Beads"

_foods, 2023, doi:10.3390/foods12173272_

Round 1

Reviewer 1 Report

The authors of the submitted manuscript aimed to improve the stability and controlled release of C-phycocyanin under in-vitro simulated gastrointestinal conditions. They evaluated the use of Ionic gelation beads made of alginate-agavins-polysaccharides for this purpose. The experiments were well-designed and conducted systematically. The study design and results presentation are clear, but the discussion section of the article is weak in some parts. There are many language mistakes, both grammatically and stylistically. The English and style require careful reorganization. In general, the manuscript is interesting and has enough novelty. However, some issues must be clarified.

As a reviewer I have the following comments:

- I recommend that you change the title and use a clearer title. The current title does not state the main topics of the study.

- The abstract should be revised and restructured.

- The authors should present in the abstract the major findings of this study in order to make the article attractive and induce reader's interest to read the full text.

-Keywords: It is better to write the keywords in alphabetical order.

- Authors should provide a literature review of previous research on the same topic, and highlight the novelty of the current manuscript.

- The study's objectives should be more clearly stated.

- The conclusions should be integrated with more detailed results summarizing all the study and must reflect the innovation of this study and the perspectives.

- Please rewrite the conclusion section (Lines 512-524).

The English and style require careful reorganization.

Author Response

Dear Reviewer, 1.

Thank you for your valuable comments and suggestions throughout the manuscript ID: FOODS 2535945. We are also grateful for the opportunity to be a part of Foods MDPI publications. We hope that our responses were agree with the ones requested by the journal. We have made every effort to respond promptly. We are attaching the list of answers in the following table, and We yellow highlighted all changes in the text.

Best regards

PhD. Brenda Hildeliza Camacho Díaz

Reviewer 2 Report

Journal Foods (ISSN 2304-8158)

Manuscript ID: foods-2535945

Type: Article

Title: Ionic gelation beads on alginate-agavins-polysaccharides to improve stability and controlled release for c-phycocyanin under in-vitro simulated gastrointestinal conditions.

Authors:

Alejandro Londoño Moreno , Zayra Mundo Franco , Margarita Franco Colin , Carolina Buitrago Arias , Martha Lucía Arenas Ocampo , Antonio Ruperto Jiménez Aparicio , Edgar Cano Europa * , Brenda Hildeliza Camacho Díaz *

General Comment

The general idea and the concept is interesting.

However, it was quite difficult for me to follow the text. Most likely because much information was given, and it was quite condensed. For the authors who have carried out the research the understanding of the experimental process comes firsthand. For me I was not always easy to follow the text, and not because of the quality of the English, but due to the way that the work is presented. Maybe the authors should go back to the text, work on it to make it more understandable.

In details:

1.    I would like to be clearer the description of the experimental procedure and specifically section 2.2.

2.    The authors use the terms: «Retention and diffusion capacity». It would be better to give the definitions at the beginning of the article.

3.    How many replications were done for the statistical processing of the results?

Line by line:

Line 84: « …. precipitated with a saturated solution of (NH4)2SO4 at 4 °C 83 for 24 h in complete darkness, then subjected to dialysis and lyophilized».

All experiments and sample handling performed in total darkness?

Line 139: I would like the authors to explain what they mean by the term “stability test” and why they think there were changes of CPC concentration, after 120 min (line 142 – «………after the encapsulation process»).

Line 188 -190:  This sentence should be further analyzed. Why this equation was chosen, and what does the terms k and n means? And yet were this equation was used ?

Lines 218 – 219 and lines 245 – 247.

In lines 2018 – 219 is stated «…… meaning that higher amounts of agavins resulted in lower efficiency. On the other hand, CMC did not seem to have a significant influence on the encapsulation efficiency».

In lines 245 – 247 is stated «Finally, matrix M4, with carboxymethyl cellulose and the aforementioned polysaccharides, achieved an encapsulation efficiency above 98% when ……»

Could the authors explain this “contradiction”?

Line 263: The authors performed FT-IR and found changes in the secondary structure of the protein during encapsulation (indicating changes in the secondary structure of C-phycocyanin). After phycocyanin release, this structure alteration still existed and if so, did it affect its functional properties?

Line 280: «Within the design of the encapsulation, release, and dosing system for CPC…..»

What does dosing system mean?

Figures

Fig 2:   Perhaps it would be better the author provide in a uniform way the results of this Figure; either as a Table or as a Plot-bar.

Yet, and most importantly, I would like the authors to make statistical comparisons not only against M1 but also between M1, M2, M3 and M4 (Each bar represents mean ± SEM. (✱) p < 0.05 compared to the control group M1).

Also, the lines 253 – 257 better incorporated in the text.

Fig 4: On the left side of the figure is the % retention and on the left side is the % diffusion for the all the 4 combinations tested. However, there are no corresponding gradations between the left and right sides, e.g. between D and H. Can the authors provide an explanation for this?

Fig 6:  What accounts for the decrease in CPC in all samples during the SGP phase?

For the same Figure, why is there such a significant difference between phycocyanin concentration (mg) and luminocity? Is there any correlation graph between concentration and luminocity?

Author Response

Dear Reviewer, 2.

Thank you for all suggestions throughout the manuscript and we appreciate for the opportunity to be part of a MDPI-FOODS Manuscript ID (foods-2535945). We attended all your indications as soon as possible. We hope that the answers were adequate in the corrected manuscript. All the suggestions are highlighted in yellow in the text.

Best regards.

PhD. Brenda Hildeliza Camacho Díaz

Corresponding author.

Reviewer 3 Report

I have reviewed Manuscript entitled " Ionic gelation beads on alginate-agavins-polysaccharides to im- 2 prove stability and controlled release for c-phycocyanin under 3 in-vitro simulated gastrointestinal conditions". Manuscript is written well and nicely presented. However, in order to improve overall quality of work following suggestions must be incorporated in final version of manuscript:

1) In Abstract section incorporate significant results of each parameter that is performed in this work in a logical manner.

2) Incorporate significance of alginates in terms of controlled release potentials in Introduction section.

3) In addition to Korsmeyer–Peppas model apply zero order and first order kinetic models too on release data?

4) What are the results of value of n with respect to Korsmeyer Peppas Model?

5) Figure 3 correct x-axis and y-axis title's formatting. Units must be in small brackets and so so 

6) Rate constant for Korsmeyer Peppas Model must be written as Kkp not simply as K everywhere?

7) Figure 7 add images with Magnification scale on each image.

Thank You

Minor English correction required in terms of sentence structure.

Author Response

Dear Reviewer, 3.

Thank you for your comments and suggestions throughout the manuscript Manuscript ID (foods-2535945) and we appreciate the opportunity to be part of a MDPI-FOODS. We attended all your indications as soon as possible. We hope that the answers were adequate in the corrected manuscript. All the suggestions are yellow highlighted in the text.

Kind regards

PhD. Brenda Hildeliza Camacho Díaz

Corresponding author.

Reviewer 4 Report

A. Overview

1. In this manuscript, the authors present an experimental study on improve c-phycocyanin stability and control its release under in-vitro simulated gastrointestinal conditions by alginate-agavins-polysaccharides

2. The contents are expressed clearly; the manuscript is well organized.

3. The authors have acknowledged recent research on this topic.

4. Keywords: Keywords should be different from the words included in the title

B. Detailed analysis.

Abstract: Be clear, and objective. State briefly what you did, how did you do it, the quantitative results you, and state the novelty of your work.

1. Introduction: provides an interesting approach to the subject.

2. Materials and Methods:

a detailed description of the statistical techniques used are given.

– The details of sampling and determination are clearly stated in this work.

2.7. In vitro digestion of matrix-type encapsulates: While the authors mentioned following the INFOGEST protocol for in vitro digestion, it would be beneficial to provide a brief summary.

3. Results:

The results section appears to be clear and well-presented, highlighting the encapsulation efficiency of different matrix-type encapsulates with varying wall materials and CPC concentrations.

- Explicitly state the number of replicates performed for each measurement and provide measures of variability to assess the robustness and reliability of the reported encapsulation efficiency values.

Author Response

Dear Reviewer, 4.

Thank you for your valuable comments and suggestions throughout the manuscript ID (foods-2535945) and we appreciate the opportunity to be part of a MDPI-FOODS. We tried to attend all your indications as soon as posible. We hope that the answers were adequate in the corrected manuscript. All the suggestions are yellow highlighted in the text.

Kind regards

PhD. Brenda Hildeliza Camacho Díaz
